# Lysozyme Fibrils Alter the Mechanism of Insulin Amyloid Aggregation

**DOI:** 10.3390/ijms22041775

**Published:** 2021-02-10

**Authors:** Mantas Ziaunys, Andrius Sakalauskas, Tomas Sneideris, Vytautas Smirnovas

**Affiliations:** 1Life Sciences Center, Institute of Biotechnology, Vilnius University, LT-10257 Vilnius, Lithuania; mantas.ziaunys@gmail.com (M.Z.); sakalauskas.and@gmail.com (A.S.); sneideris.t@gmail.com (T.S.); 2Department of Chemistry, University of Cambridge, Cambridge CB2 1EW, UK

**Keywords:** insulin fibrils, lysozyme fibrils, amyloid aggregation, aggregation mechanism

## Abstract

Protein aggregation into amyloid fibrils is linked to multiple disorders. The understanding of how natively non-harmful proteins convert to these highly cytotoxic amyloid aggregates is still not sufficient, with new ideas and hypotheses being presented each year. Recently it has been shown that more than one type of protein aggregates may co-exist in the affected tissue of patients suffering from amyloid-related disorders, sparking the idea that amyloid aggregates formed by one protein may induce another protein’s fibrillization. In this work, we examine the effect that lysozyme fibrils have on insulin amyloid aggregation. We show that not only do lysozyme fibrils affect insulin nucleation, but they also alter the mechanism of its aggregation.

## 1. Introduction

The event of protein or peptide aggregation into amyloid fibrils is considered to be the cause of multiple disorders [1,2,3]. Despite significant progress towards understanding how such insoluble and cytotoxic aggregates form, there is still no clearly defined mechanism of fibrillization, with new ideas or hypotheses appearing every year [4,5,6]. This, in turn, has made it extremely difficult to find an effective treatment or cure, as countless potential anti-amyloid compounds have failed to pass clinical trials [7,8]. Because these disorders affect millions of people worldwide and the number of cases is expected to increase further [9,10], it is important to obtain a better comprehension of the overall mechanism of protein amyloid aggregation.

Currently, the entire process can be broken down into primary nucleation, elongation, and secondary nucleation. First, the conformational change of a protein’s structure, leading to the formation of a stable primary aggregation center (nucleus) [11]. This is both the critical step in launching a cascade of protein aggregation, as well as the slowest step [12], which requires a plethora of events and factors to proceed in just the right way to achieve a stable nucleus [11]. Afterward, these primary nuclei elongate [13] by templating their structure [14] onto nearby homologous protein molecules and incorporating them into the aggregate’s structure. At later stages of aggregation, secondary processes begin to occur, which include fibril fragmentation and surface-catalyzed nucleation. Once fibrils reach a critical length, they begin to fragment and create new fibril ends [15], which are also capable of elongating. Surface-catalyzed nucleation (secondary nucleation) is a process during which native protein molecules can aggregate into nuclei on the surface of fibrils, using it as a catalyst for their formation [16]. Additional secondary events have also been proposed, which take place during aggregation, such as fibril ends of one type of amyloid serving as templates for partial misfolding of other proteins [17].

In recent years, there have been multiple reports indicating a cross-interaction between amyloid proteins [18,19,20]. Taking into consideration how unlikely it is for two different amyloid-related disorders to occur at the same time, it was hypothesized that the formation of a particular protein’s amyloid fibrils may be responsible for the appearance of a different protein’s aggregates [21]. Since fibril elongation requires the aggregate and native protein to have either exact or very similar amino acid sequences [22,23], this process can be ruled out for such cases as amyloid-beta fibrils (associated with Alzheimer’s disease) enhancing the aggregation process of alpha-synuclein (associated with Parkinson’s disease) [24]. This leaves the only other available options—fibril-surface catalyzed nucleation (i.e., secondary nucleation) or fibril ends inducing a partial protein misfolding. It is still unclear whether these processes template the fibril’s conformation onto the nuclei or if they act as catalysts and the conformation of the resulting aggregates is dictated by the environmental conditions [25,26,27]. In the case of heterogenous seeding, the second option seems to be the most probable.

There have been several studies displaying heterogenous cross-interactions of neurodegenerative disease-related amyloid proteins. It was shown that amyloid-beta aggregates are capable of inducing/enhancing the fibrilization of non-aggregated alpha-synuclein [24], Tau [28], and prion proteins [29]. Similarly, alpha-synuclein aggregates had a positive effect on Tau [30] and prion protein fibrilization [31]. It was even shown that these cross-interactions may lead to the inhibition of aggregation, as was the case with prion proteins and amyloid-beta [32]. Considering that these interactions were between neurodegenerative disease-related proteins, which shared a localization in vivo, it was interesting to examine whether completely unrelated amyloid proteins could also have an effect on each other’s aggregation. For this reason, two model amyloidogenic proteins were chosen—human insulin [33] and hen egg-white lysozyme [34]. Insulin is an ideal candidate to gather mechanistic insight into the process, as it has the ability to form fibrils with distinct conformations and replication properties at different conditions, such as pH value [35] or protein concentration [26]. The aggregation conditions where such conformation variations are visible are far from physiological (low pH value, 20% acetic acid solution, 60 °C), however, they are necessary in order to obtain monomeric insulin [36] and have adequate repeatability between experiments [26]. Thus, if lysozyme fibrils have any effect on insulin aggregation, it will clearly reflect in the aggregation times, fibril secondary structure, and morphology, as well as the fluorescence intensity of an amyloid-specific fluorescent dye—thioflavin-T (ThT) [37,38]. In addition, both protein molecules contain no similarities between their amino acid sequences, negating any possible elongation events.

In this work, we show that not only do lysozyme fibrils increase the rate of insulin nucleation, but they also have a peculiar effect on the actual mechanism of its aggregation. At large concentrations, lysozyme fibrils induced a similar effect as a high concentration of insulin protein under the tested conditions. In addition, when the initial solution contained both types of fibrils, they experienced a synergistic effect, altering the aggregation pathway.

## 2. Results

During unseeded aggregation, in both cases where lysozyme fibrils (further referred to as LF) are present (Figure 1B,C), there is a noticeable initial decrease in signal intensity (Appendix A), caused by the sample reaching 60 °C and thus reducing the fluorescence intensity of LF-bound ThT. This was followed by a slow, continuous increase in signal intensity (observed in the control samples), which could be the result of LF gradually settling at the bottom of the wells, where the fluorescence intensity was measured. In order to account for this effect on the overall aggregation kinetics, the control sample signal intensity was subtracted from the reaction solution signal.

The average half-time (t_50_) value of spontaneous insulin aggregation reaction was ~300 min (Figure 1A,F). The addition of 2 µM LF (in this case and further throughout the manuscript, fibril concentration refers to the concentration of protein monomers in their aggregated state) reduced the t_50_ value to ~180 min (Figure 1B,F) and increased the normalized curve slope value (Figure 1E), indicating a positive effect on the rate of nucleation and elongation. However, the increase in the slope value was less substantial than the change in t_50_ and may be within the margin of error, as observed in additional repeats of the experiment (Appendix A). When 200 µM LF was present in the sample (Figure 1C), t_50_ was similar to conditions when no LF were present (~300 min), and there was a decrease in the slope value (Figure 1E,F). In this case, there was also a massive shift in the insulin-fibril-related ThT fluorescence intensity when compared to the other two conditions (Figure 1D). Such a change in ThT fluorescence intensity was an indicator of the formation of a different insulin fibril conformation under AC conditions (20% acetic acid solution, containing 100 mM NaCl) [26]. Additional repeats of unseeded aggregation (Appendix A) presented results with similar tendencies.

In order to test whether fibril ends or their surface was responsible for the change in t_50_ and bound-ThT fluorescence intensity, insulin aggregation was carried out in the presence of 2 µM sonicated LF and compared against aggregation with non-sonicated LF (Appendix A). The nearly identical t_50_ and slope values, as well as bound-ThT fluorescence intensities, indicated that the number of LF ends did not dictate the rate of insulin nucleation or elongation. The effect of intermediate LF concentrations was also examined (Appendix A), however, the signal from LF-bound ThT had a very significant effect on the overall signal intensity, as the high bound-ThT intensity insulin conformation only appeared at high LF concentrations.

Seeing as the presence of LF had such a significant effect on insulin fibril formation, and the ThT fluorescence intensity hints at the formation of a different aggregate conformation, atomic force microscopy (AFM) images were acquired to examine if there were any changes to fibril morphology. When there was a small concentration of LF added to the sample, there were no significant changes observed (Figure 2A,C), and average fibril height and width remained even (5 nm and 27 nm, respectively). When 200 µM of LF was present, it became extremely difficult to differentiate between insulin and lysozyme fibrils, as there was a high propensity towards cluster formation (Figure 2B,D). One aspect worth noting was that when insulin was aggregated with 200 µM LF, the clusters were significantly more massive than in the control LF samples (Figure 2B,D). Due to such cluster formation, it was impossible to draw any conclusions about the effect LF had on insulin fibril morphology (single fibril width, height, and length could not be accurately measured), which requires an alternative method to determine whether there were any LF-induced structural changes.

To determine if there were any structural differences between insulin fibrils formed with and without LF, each sample’s FTIR spectra were scanned (Figure 3A). Because of the existence of both types of aggregates, the lysozyme fibril spectrum was subtracted from the insulin with LF spectra as described in the Materials and Methods Section. The most noticeable distinction, which was observed in the second derivatives of FTIR spectra (Figure 3), was the additional band at 1620 cm^−1^, which was present only in the case when insulin was aggregated in the presence of 200 µM LF. As previously reported [26], such a band appears only in the FTIR spectra of insulin fibrils that have spontaneously formed at high initial protein concentrations (1.0 mM, termed high concentration fibrils—HCF) in AC solution. There was also a shift in intensity at 1641 cm^−1^, a change which was also associated with the formation of HCF. This indicates that 200 µM of LF alters the aggregation mechanism, leading to a different insulin fibril secondary structure, while 2 µM of LF has no such effect. Two additional repeats of FTIR spectra were scanned for samples prepared from different batches and similar tendencies were observed (Appendix A).

In order to account for the possibility that the formation of HCF iwas caused by a crowding effect, rather than LF themselves, insulin aggregation was carried out in the presence of 200 µM lysozyme monomers (Appendix A). The presence of lysozyme monomers reduced the t_50_ value, similarly as 2 µM LF, however, they did not cause such an increase in ThT fluorescence intensity or changes to insulin fibril secondary structure (Appendix A) as was the case with 200 µM LF.

As lysozyme fibrils appeared to have an effect on nucleation, as well as the aggregation curve slope value (possible faster rate of elongation), seeded aggregation experiments were carried out in the absence or presence of 2 µM or 200 µM LF (Figure 4, Appendix A). When seeded aggregation occurred in the AC solution without any lysozyme fibrils, we observed typical seed-concentration-dependent kinetics (Figure 4A). The same was true when there was 2 µM of LF added to the solution (Figure 4B). In both cases, the overall ThT fluorescence intensity remained relatively low, however, there was a slightly higher end-point intensity when there was a large concentration of insulin seed added (Figure 4A,B,F). The average t_50_ values of samples without LF were higher throughout the entire seed concentration range when compared to the 2 µM LF samples, and both followed a linear trend, which converged at higher seed concentrations (Figure 4E). The slope values were also slightly higher in the case of 2 µM LF, which was observed in the spontaneous aggregation experiment as well (Figure 1E).

However, when the samples contained 200 µM of LF, we could no longer see a clear trend (Figure 4C). When there was a lot of insulin seed present in the solution, the resulting ThT fluorescence intensity was much higher than in the case of 0 or 2 µM LF (Figure 4F). This intensity value then began to drop until 10^−3^–10^−4^% insulin fibril concentration, nearly reaching the intensity of both other conditions. Afterward, it began to increase again, as nucleation events became dominant and the different fibril conformation displayed higher ThT fluorescence intensity, formed, as seen in the spontaneous aggregation experiment (Figure 1D). While there were no major differences observed in the aggregation curve slope values (Figure 4D), there was a noticeable distinction in the t_50_ values (Figure 4E). They were higher than in both other conditions at low initial insulin seed concentrations and converged in the 10^−5^–1% range. This t_50_ value dependence on initial seed concentration obtained a sigmoidal shape, rather than being linear, as seen in both the 0 µM and 2 µM LF conditions, respectively. Since this divergence appeared only when the initial insulin fibril concentration was low (10^−5^%), it is likely that LF did not have a prominent effect on elongation, as in such a case, we would observe lower t_50_ values throughout the entire seed concentration range. Additional repeats of seeded aggregation (Appendix A) displayed similar tendencies.

In order to determine the cause of the high variation in ThT fluorescence intensity during seeded aggregation, FTIR spectra of insulin fibrils, formed in the absence or presence of 200 µM LF at high (Figure 5A,D), intermediate (Figure 5B,E), and low (Figure 5C,F) initial seed concentrations were examined. When the initial insulin fibril concentration was 1%, the FTIR spectra of formed fibrils were similar and did not provide a clear indication of the existence of a different type of aggregate. When the seed concentration was 10^−4^%, the HCF-related minima at 1620 cm^−1^ appeared in the sample with 200 µM LF, however, it was not as expressed as in the case of HCF, which means the sample likely contained both types of aggregates. When the initial seed concentration was very low, the insulin fibrils formed with 200 µM have a FTIR spectrum that was similar to the one seen in unseeded aggregation, which was to be expected with such a low initial fibril concentration.

The similarities between high initial seed concentration sample FTIR spectra (with minor variances due to sampling light scattering) did not explain why the resulting ThT fluorescence intensity was so high (Figure 4F) when such a large concentration of seed should effectively replicate its structure and not provide enough time for HCF nuclei to form. A high number of initial aggregation centers and low t_50_ value both led to fewer nucleation events, meaning the existing nuclei each had a larger portion of available monomers. This would allow the formation of considerably longer fibrils than in the case of 10^−4^% initial seed concentration. These longer fibrils could potentially bind more ThT molecules, as well as have a stronger interaction with LF, thus resulting in a higher bound or trapped ThT fluorescence intensity. In order to test this hypothesis, the high, intermediate and low initial fibril samples, each containing 200 µM LF, were scanned before and after a brief round of sonication (Appendix A). On average, the 1% and 10^−8^% samples experienced a larger reduction in ThT intensity than the 10^−4^% sample, however, the change percentage values were within the margin of error, and the conclusion that fibril length was solely responsible for this effect cannot be confirmed.

## 3. Discussion

Considering that previously reported cases analyzed the interaction of amyloid proteins that had the ability of coming into contact in vivo (alpha-synuclein, Tau, prion proteins, and amyloid-beta) [24,28,29,30,31,32], it was interesting to examine if such amyloid cross-interactions can occur between proteins that do not share a localization. From these results, it is clear that the presence of lysozyme fibrils has a major impact on the aggregation mechanism of insulin. If we compare the fibrillization kinetics of both spontaneous and seeded aggregation (at extremely low initial seed concentrations) in the absence and presence of 2 µM LF, they enhance the rate of nuclei formation. Such an aggregation-promoting effect, caused by amyloid fibrils, was also observed in the case of amyloid-beta [28], as well as alpha-synuclein [30,31]. This could be achieved by either the surface of lysozyme fibrils acting as a catalyst for insulin nucleation events (fibril ends causing a partial misfolding of insulin can be ruled out, as a change in their number did not alter aggregation) or by creating an interface between the solution and its hydrophobic surface (interface-enhanced aggregation [39]). However, both of these possible mechanisms are only viable at comparatively low LF concentrations, as we observe the appearance of other mechanism-altering effects at a high LF concentration.

The most interesting aspect of this heterogenous interaction is the ability of lysozyme fibrils to induce the formation of a different insulin fibril conformation, an event that has not been observed in previous studies. We have recently reported that when insulin aggregates in a 20% acetic acid solution, the protein’s initial concentration plays a crucial role in determining what type of aggregates are formed [26]. If the initial concentration is relatively low (0.2 mM, named low concentration fibrils—LCF), then the resulting fibrils are shorter, have a lower tendency towards self-association, and have a low bound-ThT fluorescence intensity. Conversely, when the initial protein concentration is 1.0 mM, the resulting fibrils are longer, have a higher tendency towards aggregate clump formation, and have a high bound-ThT fluorescence intensity (high concentration fibrils—HCF). Both of these conformations also possess differences in their FTIR spectra, with the most notable being a band at 1620 cm^−1^ in the case of HCF. When there are 200 µM of LF present, we observe the appearance of the HCF-like conformation, even though the concentration of insulin is 0.2 mM in all cases and should result in a LCF conformation. One possibility is that if insulin molecules associate with the surface of lysozyme fibrils, then this event could create a localized high-concentration area and facilitate the formation of the HCF conformation. Such a protein-condensing event was also reported to enhance the elongation of prion proteins [40]. This would also explain the previously mentioned effect on nucleation. If the effective concentration of insulin is high at certain areas, then the possibility of nucleation events would also be enhanced.

Seeded nucleation progresses without any abnormalities in the absence or presence of 2 µM LF, with LF reducing the t_50_ value and increasing the curve slope value, as was the case with spontaneous aggregation. However, once 200 µM of LF are added, it leads to extremely unusual aggregation kinetics. High initial insulin LCF seed concentrations result in a relatively high ThT fluorescence, which is peculiar, as such a high concentration of LCF insulin fibrils should replicate their structure and result in a low fluorescence intensity. The effective replication of LCF was also confirmed by FTIR, which indicated that other factors were involved in the high ThT fluorescence. Sample sonication reduced their fluorescence intensity, however, the proportional decrease was not significant enough to indicate that fibril length is the sole reason for higher fluorescence intensity and other, more complex factors may be involved. Another peculiar factor is the sigmoidal shape of the t_50_ value dependence on initial seed concentration when insulin is aggregated in the presence of 200 µM LF. It seems that LF only has a prominent effect on nucleation, as the t_50_ values converge at 10^−4^%. However, this effect may be more complex than LF simply acting as an interface for nucleation. Considering that 2 µM LF reduces the t_50_ value by roughly 100 min, we would expect 200 µM LF to decrease it even further, but this is not the case. 200 µM LF actually slows down nucleation to a point where t_50_ values reach the ones observed in samples with no LF present. A possible explanation is that LF force the generation of HCF fibrils, which, in turn, have a lower rate of self-replication, thus increasing the reaction’s t_50_ value. This lower rate of elongation is seen in lower slope values for both spontaneous nucleation and seeded aggregation with low initial fibril concentrations in the presence of 200 µM LF.

Taking all of these results together, it appears that lysozyme fibrils can alter the mechanism of insulin aggregation in more ways than one. The enhanced rate of nucleation, formation of a new fibril conformation, and changes to the overall aggregation kinetics all appear to point towards the idea that lysozyme fibrils are capable of condensing non-aggregated insulin and altering the environmental conditions, which favor the formation or propagation of a different insulin fibril conformation. Such synergy between two highly distinct proteins under non-physiological conditions may be an indicator that these cross-interactions could be a generic feature of amyloid proteins.

## 4. Materials and Methods

### 4.1. Fibril Preparation

Human recombinant insulin powder (Sigma-Aldrich, St. Louis, MO, USA, cat. No. 91077C) was dissolved in a 20% acetic acid solution, containing 100 mM NaCl (further referred to as AC) to a final protein concentration of 200 µM (ε_280_ = 6335 M^−1^cm^−1^, M = 5808 Da). The resulting solution was distributed to 1.5 mL test-tubes (1 mL each) and incubated at 60 °C for 24 h under quiescent conditions.

Hen egg-white lysozyme powder (Sigma-Aldrich, St. Louis, MO, USA, cat. No. L6876) was dissolved in a 50 mM sodium phosphate buffer (pH 6.0) containing 2 M guanidine hydrochloride (GuHCl) to a final protein concentration of 200 µM (ε_280_ = 37,970 M^−1^cm^−1^, M = 14,313 Da). The resulting solution was distributed to 1.5 mL test-tubes (1 mL each), each containing 2 3 mm glass-beads and incubated at 60 °C for 72 h under 600 rpm agitation. The resulting fibril solutions were centrifuged at 10,000× *g* for 30 min, the supernatant was removed, and the fibril pellet was resuspended into the AC solution. This centrifugation and resuspension procedure was repeated 4 times. Finally, the fibrils were resuspended into 400 µL AC solution in each test-tube and combined to yield a solution with 500 µM fibril concentration (fibril concentration is the concentration of monomers in their aggregated state). This solution was incubated for an additional 24 h at 60 °C without agitation in order to negate any possible changes the different environmental conditions may have on the fibril structure during the following aggregation experiments. To reduce the possibility of reaggregation events, the fibrils were not sonicated before use.

### 4.2. Aggregation Kinetics

Human recombinant insulin powder was dissolved as previously described to a final concentration of 500 µM. Then, the insulin stock solution was mixed with a thioflavin-T (ThT) stock solution (10 mM in MilliQ water), lysozyme fibril solution, and AC solution to a final mixture containing 200 µM insulin, 100 µM ThT, and 0, 2, or 200 µM lysozyme fibrils. These mixtures were distributed to low-protein binding 96-well plates (100 µL each well, 3 repeats of every solution). Aggregation kinetics were tracked by monitoring ThT fluorescence using 440 nm excitation and 480 emission wavelengths in a ClarioStar Plus (BMG Labtech, Ortenberg, Germany) platereader at 60 °C with no agitation between readings.

For seeded aggregation experiments, the insulin fibril solution was sonicated for 10 min using a Bandelin Sonopuls (Berlin, Germany) ultrasonic homogenizer (MS73 tip, 40% maximum power, 30 s sonication/30 s rest intervals). All the solutions for seeded aggregation were prepared as described previously, additionally containing a range of insulin fibril concentrations (from 1% to 10^−8^% of total protein concentration). Control samples, only containing 2 or 200 µM of lysozyme fibril concentrations were monitored identically to subtract their signal from the combined insulin-lysozyme solution signal.

Aggregation half-time (t_50_) and kinetic curve slope values were calculated by applying a linear fit to the data points ranging from 40% to 60% of normalized intensity values and interpolating the time at which 50% of intensity was reached. In cases when the high initial lysozyme fibril concentration resulted in notable fluctuations at the beginning of the reaction (ThT fluorescence reduction due to the sample reaching 60 °C), normalization was done by omitting the initial 25 min data points. The insulin-fibril-bound ThT fluorescence intensity values were calculated by subtracting the fluorescence intensity at the beginning of the reaction from the signal intensity at the end of the reaction (for these calculations, kinetic curves with subtracted control intensities were used). Raw kinetic data is available as Appendix A.

### 4.3. Atomic Force Microscopy

Aliquots from spontaneous aggregation fibril samples were diluted 4 times using the AC solution. The atomic force microscopy (AFM) images were acquired as described previously [41]. Briefly, 30 µL of each solution was distributed on freshly-cleaved mica, incubated for 1 min, gently washed with MilliQ water, and air-dried. 1024 × 1024 resolution AFM images were acquired using a Dimension Icon (Bruker, Billerica, MA, USA) atomic force microscope in tapping mode (3 images for every condition). The AFM images were flattened using Gwyddion 2.5.5 software. Fibril height and width were determined by tracing perpendicular to the fibril’s axis (only non-clumped fibrils or fibril ends were used in order to acquire accurate, single-fibril data). A total of 200 traces were obtained from all 3 AFM images for each sample.

### 4.4. Fourier-Transform Infrared Spectroscopy

FTIR spectra of each aggregation sample were acquired as described previously [41]. Briefly, the fibril solutions were centrifuged for 30 min at 10,000× *g*, then the supernatant was removed, and the fibril pellets were resuspended in 500 µL D_2_O, containing 200 mM NaCl (higher ionic strength result in higher fibril self-association and improve sedimentation [42]. This centrifugation and resuspension procedure was repeated 4 times. Finally, the fibrils were resuspended in 150 µL of D_2_O (without NaCl), sonicated for 30 s (MS72 tip, 20% power). For each sample, 256 interferograms were obtained using a Vertex 80v (Bruker, Billerica, MA, USA) IR spectrometer at room temperature under near-vacuum conditions. FTIR spectra were analyzed using GRAMS software. In each case, both the D_2_O and the lysozyme fibril (where necessary) spectra were subtracted. Lysozyme fibril spectrum subtraction factor was based on the portion of peptide bonds related to each protein (factor was set to 0.72 for samples with 200 µM LF and 0.025 for 2 µM LF). Raw FTIR data is available as Appendix A.

## Figures and Tables

**Figure 1 ijms-22-01775-f001:**
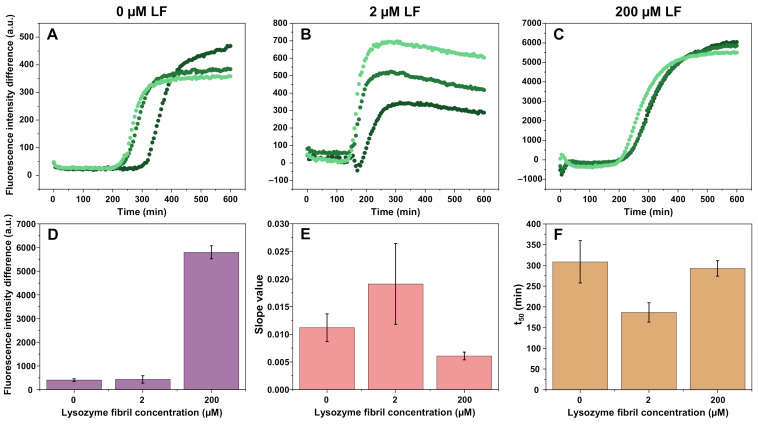
Aggregation of insulin (200 µM) in AC solution (20% acetic acid solution, containing 100 mM NaCl) with 0 µM (**A**), 2 µM (**B**), and 200 µM (**C**) lysozyme fibrils (control sample signal intensities are subtracted). The fluorescence intensity of ThT bound to insulin fibrils (**D**), normalized curve slope values (**E**), and aggregation half-time (t_50_) values (**F**). Fluorescence intensity differences, slope values, and t_50_ values were determined after subtracting the signal intensities of the control sample from the reaction sample. Different shades of green kinetic curves indicate three separate repeats.

**Figure 2 ijms-22-01775-f002:**
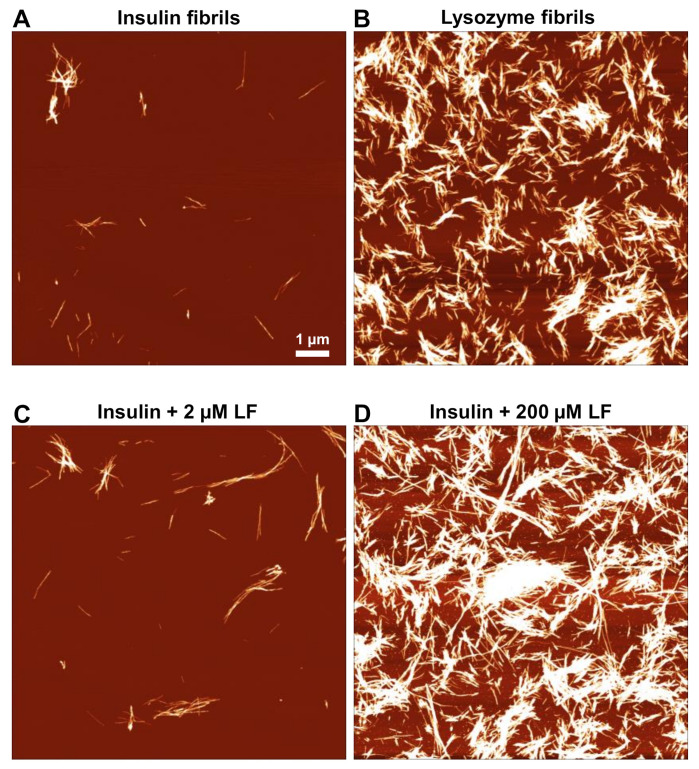
Atomic force microscopy (AFM) images of insulin (**A**) and lysozyme (**B**) fibril control samples and insulin aggregated in the presence of 2 µM (**C**) and 200 µM (**D**) lysozyme fibrils.

**Figure 3 ijms-22-01775-f003:**
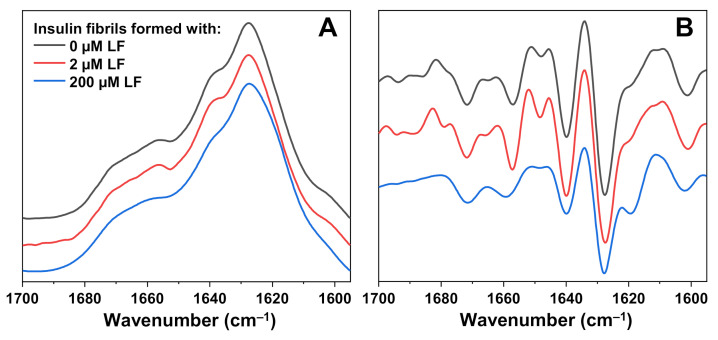
FTIR spectra of insulin aggregated in AC solution (20% acetic acid solution, containing 100 mM NaCl) in the absence or presence of 2 µM and 200 µM lysozyme fibrils (**A**) and their second derivatives (**B**). In cases when lysozyme fibrils were present in solution, their corresponding spectrum was subtracted from the mixture’s spectrum as described in the Section Materials and Methods.

**Figure 4 ijms-22-01775-f004:**
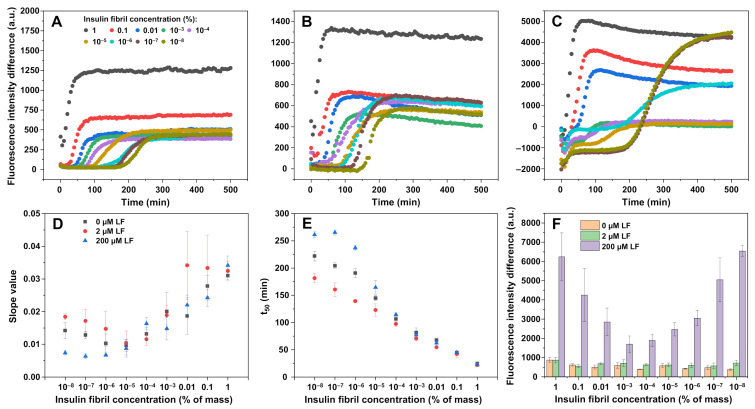
Insulin (200 µM) seeded aggregation kinetics in the absence (**A**) and presence of 2 µM (**B**) and 200 µM (**C**) lysozyme fibrils (control sample signal intensities are subtracted). Normalized curve slope values (**D**), aggregation half-time (t_50_) values (**E**), and fluorescence intensity of ThT bound to insulin fibrils (**F**). Fluorescence intensity differences, slope values, and t_50_ values were determined after subtracting the signal intensities of the control sample from the reaction sample. A representative curve is shown for every condition, additional repeats are available as Appendix A.

**Figure 5 ijms-22-01775-f005:**
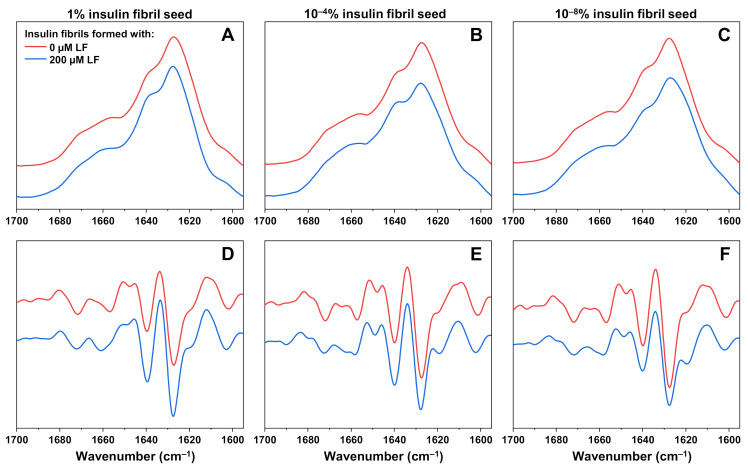
FTIR spectra and their second derivatives of insulin fibrils prepared in the absence or presence of 200 µM LF, using an initial insulin fibril seed concentration of 1% (**A**,**D**), 10^−4^% (**B**,**E**), and 10^−8^% (**C**,**F**).

## Data Availability

All presented data is available as Appendix A.

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
