# Peer review of "Lysozyme Fibrils Alter the Mechanism of Insulin Amyloid Aggregation"

_ijms, 2021, doi:10.3390/ijms22041775_

Round 1

Reviewer 1 Report

Ziaunys et al. showed that lysozyme fibrils may affect insulin aggregation. They firstly performed thioflavin-T assay and then examined in detail by using atomic force microscopy and Fourier-transform infrared spectroscopy. The method was sophisticated. However, this manuscript is highly descriptive. In addition, it is difficult to interpret the linkage between the current results and neurodegenerative disorders, although the authors mentioned abnormal protein aggregation in relation to neurodegenerative disorders.

  1. Why insulin and lysozyme fibrils were selected in this experiment? The hypothesis of the authors is probably that there is a heterogenous interaction between different amyloidogenic proteins in neurodegenerative disorders such as Alzheimer’s and Parkinson’s disease. If so, β-amyloid or α-synuclein should be used in the experiments.

  1. In the introduction, the authors describe that type-II diabetes is included in neurodegenerative disorders. Even if type-II diabetes is a protein misfolding disease, it is not a neurodegenerative disorder.

  1. I do not understand that insulin is pathologically aggregated in the brain. The authors should clarify the reason why aggregation of insulin was evaluated in this study. If insulin aggregation is important in neurodegenerative disorders, in vivo experiments should be added.

  1. The authors failed to show dose-depending effects of lysozyme fibrils on insulin aggregation in Figure 1 and 2. It is needed to examine the effects of around 20 μM of lysozyme fibrils in these experiments.

  1. In the beginning of the result section, there is only one subheading: “Unseeded aggregation”. Please check it.

  1. In discussion, only one reference is quoted. The authors should discuss the relationship between presented results and previous studies.

Author Response

Please find comments in the attached file.

Reviewer 2 Report

Understanding the mechanisms of protein aggregation into amyloid fibrils is of high interest as they are hallmarks of various neurodegenerative pathologies such as Alzheimer, Parkinson diseases and type II diabetes. In this manuscript, the authors reported the study of the impact of lysozyme fibrils (LF) presence on insulin amyloid aggregation. The manuscript is pleasant to read, the context of the project is well introduced and the study is well conducted. The authors have collected numerous data of various experiments in order to understand the mechanisms originating from the insulin aggregation process modification in the presence of LF, and the discussion linked to the results is quite realistic and reasonable. However, from my point of view, the experimental conditions related to the study are not clearly addressed in the manuscript (except in the material and method section of course) and I have some concerns about it.

I am a little bit surprised than the conditions chosen for the experiments are very far from physiological conditions: aggregation at 60°C and solution in acetic acid 20%. Indeed, even if the alteration of insulin aggregation process in the presence of LF is demonstrated in these specific conditions, different interactions/processes could occur in biologically relevant conditions. This important point has to be addressed in the manuscript to avoid any confusion, as the pH, temperature and the presence of acetic acid are parameters of importance whose modification could drastically impact the aggregation process and how the presence of LF impacts insulin aggregation into fibrils. The explanation of their choice of experimental conditions and a clear mention in the manuscript that the conditions are not physiologically relevant are mandatory for transparency and a better contextualization of the study. Moreover, this could also be discussed at the end of the discussion, maybe a perspective could be to conduct similar assays in a more physiologically relevant environment?

Other main remarks:

  • Lines 268 and 286: How the fibril concentrations (of lysozyme and insulin fibrils for seeded aggregation experiments have been determined? The characterization and quantification of the fibrils is a very important point for the experiments of this study and they are not detailed enough
  • Have the authors considered to try also intermediate LF concentrations (between 2 and 200 µM) in order to have an idea of the critical LF concentration where the influence of LF changes (2µM vs 200 µM)?
  • Line 191: Concerning the “considerable drop in ThT intensity” of 1% and 10-8% (seen in Figure A4), it would be appreciated to have a percentage of decrease before and after sonication in order to compare the various conditions. Indeed, I am not so confident that the decrease in the case of 10-4% is really lower than the other conditions, it is difficult to see as the initial intensity before sonication is much lower for 10-4%., and adding the percentage could help to overcome the doubts.

Detailed remarks:

Line 28: I think than the authors wanted to say “essential / important / mandatory”” instead of “detrimental”

Line 88: Even if it is explained in the material and method section, it would be nice to explain in the text what is the abbreviation “AC” to avoid the reader to check in the first paragraph what AC means.

Caption of Figure 1 (line 90): adding the insulin concentration in the caption would help the reader to have an idea of the concentration of lysozyme fibrils compared with the insulin.

As a conclusion, when the important question of the non-physiologically relevant experimental conditions of the study will be addressed and discussed clearly in the text, I recommend a publication of this manuscript in IJMS.

Author Response

(The authors gave the same response as above.)

Reviewer 3 Report

In this work Smirnovas and co-authors report the influence of Lysozyme fibrils on amyloid aggregation of Insulin, one of the processes involved in type II diabetes.

The work is performed on a good technical level and deals with an important topic. It will be interesting for scientists studying mechanisms of amyloid fibril formation and its inhibition. I think, it will be a really nice paper if the authors will do few additional experiments to improve quality of the data and clarify the mechanism of the effect.

I strongly suggest to:

  1. To perform experiment with 0.2μM sonicated fibrils in order to clarify if the nucleation occurs on sides or ends of fibrils.
  2. Do more replicates in all aggregation experiments to have more reliable data, especially for non-seeded experiments.
  3. Improve the description of the molecular mechanism of the process.

Minor suggestions:

P1L34 Please improve the description of fibrillization mechanism: fibril breaking is one of the secondary nucleation processes and not a separate step.

P2L45 Statement "Since fibril elongation requires the aggregate and native protein to have either exact or very similar amino acid sequences [21,22] ….  This leaves the only other available process – fibril-surface catalyzed nucleation (i.e. secondary nucleation)." is not correct. Ends of fibrils of one amyloid can serve as templates for partial folding of other protein in parallel β-sheet conformation. (and they typically bind proteins with a higher affinity than fibril sides due to H-bond formation, see, for example, binding of heat shock proteins to α-synuclein).

P2L72 It is unclear why the experiments were performed at 60C. High temperature accelerates the primary nucleation. Therefore, to focus on secondary nucleation, it would be more logical to do aggregation at RT or at 37C. Moreover, the drop in ThT intensity is huge and seriously disturbs the quality of data (if to consider kinetic curves from SI)

P2L88 Please explain briefly what is "AC conditions"

Fig 1 shows significant sample-to-sample variability between 3 repeats. Please do 4-6 repeats in such cases.

Fig 1. It would be better to compare results for 0-0.2-2μM LF samples on one graph (for example show only average curves for each condition)

Fig 3. It is hard to judge whether the difference between +/- LF samples is significant without knowing how precise and reproducible are spectra. Can you provide spectra for at least 2 independently prepared +LF and –LF samples?

Figures 1 and 4. Please provide insulin concentration in the figure captions.

Figure 5. The difference in IR spectra between samples seeded with 1, 1e-4, and 1e-8 insulin fibrils (red curves in Fig 5 a,b,c) is quite visible. How do you explain this?

General comments:

 If the experiments are performed in 96-well plates, it is logical to do at least 8 repeats for each conditions in non-seeded aggregation experiments.

 The average length of formed fibrils and its dependence on the protein concentration can be used to determine the mechanism of the secondary nucleation and it is better to discuss it in more details if possible.

Author Response

(The authors gave the same response as above.)

Round 2

Reviewer 1 Report

Although the authors revised the manuscript, the rationale for this experiment is still unclear. Even if insulin has a property of amyloid, it has been probably shown neither in brain nor in other nervous system. The authors mentioned abnormal protein aggregation in relation to neurodegenerative disorders. Therefore, β-amyloid or α-synuclein should be used in the experiments.

Author Response

We have removed the emphasis on neurodegenerative disorders (Alzheimer’s disease and Parkinson’s disease) in both the abstract and introduction, as the manuscript is focused on the complex interaction between insulin and lysozyme, which are not directly related to these disorders. We hope that this clears up any possible confusion regarding the aim of the study, which is the examination of two unrelated amyloidogenic proteins.

The interaction between Amyloid-beta and alpha-synuclein has already been reported in a previous work (which is referenced in this manuscript) and such experiments would be out of scope for this particular study, which examines very different amyloidogenic proteins that do not share a localization in vivo.

Reviewer 3 Report

The authors performed requested experiments with sonicated fibrils, improved the introduction and added data for more repeats in he key figures.
Therefore, I think, the manuscript is significantly improved and can be accepted for publication.

Author Response

No response needed.

Round 3

Reviewer 1 Report

I am a researcher in the field of neurodegenerative disorders, but not in general amyloidogenic proteins. The authors have removed the emphasis on neurodegenerative disorders and just focus on the complex interaction between insulin and lysozyme. Regarding the amended manuscript, I don't feel qualified to judge about the quality of this manuscript. My previous recommendation was "reject". So, I would like to make the same decision in the current review.